# Long-Term Antibiotic Prophylaxis in Urology and High Incidence of *Clostridioides difficile* Infections in Surgical Adult Patients

**DOI:** 10.3390/microorganisms8060810

**Published:** 2020-05-28

**Authors:** Estera Jachowicz, Marta Wałaszek, Grzegorz Sulimka, Andrzej Maciejczak, Witold Zieńczuk, Damian Kołodziej, Jacek Karaś, Monika Pobiega, Jadwiga Wójkowska-Mach

**Affiliations:** 1Department of Microbiology, Faculty of Medicine, Jagiellonian University Medical College, 18 Czysta Street, 31-121 Krakow, Poland; estera.jachowicz@doctoral.uj.edu.pl (E.J.); monika.pobiega@gmail.com (M.P.); 2State Higher Vocational School, St. Luke’s Provincial Hospital, 31-100 Tarnow, Poland; mz.walaszek@gmail.com; 3St. Luke’s Provincial Hospital, 31-100 Tarnów, Poland; sulimka@lukasz.med.pl (G.S.); amac@mp.pl (A.M.); witoldz@63vp.pl (W.Z.); domator@poczta.onet.pl (D.K.); ikar.jk@gmail.com (J.K.); 4Medical Faculty, Univeristy of Rzeszów, 35-959 Rzeszów, Poland

**Keywords:** *C. difficile*, healthcare-associated infections, perioperative antimicrobial prophylaxis, antibiotic consumption

## Abstract

*Clostridioides difficile* infections are the main cause of antibiotic-related diarrhea. Most of them come in the form of healthcare-associated *Clostridioides difficile* infections (HA-CDI). The aim of the study was to analyze HA-CDI epidemiology and the relationship between antibiotic consumption and CDI epidemiology at St Luke’s Provincial Hospital in Tarnow, Poland. In 2012–2018, surveillance of CDI was carried out in adult surgical wards at St Luke’s Provincial Hospital. The data were collected in accordance with the methodology of the Healthcare-Associated Infections Surveillance Network (HAI-Net), European Centre for Disease Prevention and Control (ECDC), and the ATC/DDD system (Anatomical Therapeutic Chemical Classification System) of the World Health Organization. In total, in the study period, 51 cases of CDI involved CA-CDI (24.5%) and 147 were HA-CDIs (75.5%). The most CA-CDIs were found in the general surgery (32.6%) and urology (17.0%) wards. CA-CDI incidence was 0.7/1000 patients and for HA-CDI it was 2/1000 patients (4.4/10,000 patientdays (pds)). The highest HA-CDI incidence was in the neurosurgical departments (18/10,000 pds) and oncological surgery (8.4/10,000) pds. There was a significant positive correlation between CA-CDI and HA-CDI (correlation of 0.943, *p* < 0.001) and between the number of patients hospitalized and HA-CDI (correlation of 0.865, *p* = 0.012). The total antibiotic consumption amounted to 0.7 DDD/10,000 pds; it was the highest in the urology ward (0.84/10,000 pds) and 49.5% of the antibiotics were fluoroquinolones (0.41/10,000 pds). On the basis of regression coefficients, a positive correlation was demonstrated between the use of fluoroquinolones and the HA-CDI incidence rate. Both a high percentage of CDI cases and a high intake of antibiotics were recorded in the urology department. About half of all antibiotics were fluoroquinolones.

## 1. Introduction

Currently, *Clostridioides difficile* (CD) infections are a major challenge for modern medicine and healthcare infrastructures worldwide. CDIs are the main cause of antibiotic-related diarrhea; however, on the other hand, only 10–25% of antibiotic-related diarrhea cases confirm CD as the etiological agent of the infection [1].

The limitation of the genus *Clostridium* to *Clostridium butyricum* and related species means that a lot of existing microorganisms should not be considered *Clostridium sensu stricto*. One such example of medical significance is *C. difficile*. Based on 16S rRNA gene sequence analysis, the closest relative of *Clostridium difficile* is *Clostridium mangenotii*, and both are in the *Peptostreptococcaceae* family, which is phylogenetically far from the members of *Clostridium sensu stricto*. Based on phenotypic, chemotaxonomic, and phylogenetic analyses, a new type proposed for *Clostridium difficile* is *Clostridioides difficile* [2].

It was determined that CD binds to the components of the matrix, such as fibrinogen, collagen, vitronectin or fibronectin, and with the use of Fbp68 proteins [3], the pathogen adheres to the intestinal cells tightly, which hinders its recognition by the immune system cells. Intestinal diseases are caused only by the CD strains that produce toxins A or B. This was proven by introducing insertions into the genes *tcdA* and *tcdB* of virulent bacteria, which led to a loss of toxin production and absence of disease symptoms [4].

Empirical therapy for CDI is effective with regards to approximately 75% of cases, while 25% of patients with CDI do not respond to antibiotic therapy, and additionally, in approximately 8% of cases there are relapses. However, the fraction of relapse is impossible to estimate even with modern whole genome sequencing techniques, due to common environmental contamination [5,6].

As many as 75% of all *C. difficile* infections (CDI) are associated with hospitalization [6]. HA-CDIs (healthcare-associated *Clostridioides difficile* infections) prolong hospitalization and involve additional costs [7]. Parenteral CDI are very rare and are characterized by high mortality. In a study conducted by Urbán et al., 4129 CD strains were isolated; among which only 24 (0.58%) isolates came from parenteral sources. Eight strains were isolated from abdominal infection—appendicitis, rectal abscess, or from people with Crohn’s disease. The patients did not have diarrhea. In most cases, CD was obtained as a part of a polymicrobial flora isolated from wounds, abscesses, blood culture sample, bile sample, and a blister fluid. The isolates were most often capable of producing toxins and belonged mainly to the PCR 027 ribotype. There is an increasing number of reports of *C. difficile*-related parenteral infections [8]

Seventy percent of all CDIs are associated with hospitalization. The most significant risk factor for the development of CDI is taking antibiotics that interfere with the natural intestinal microbiota and other significant risk factors for CDI are, e.g., surgery and being over 65 years of age. Hospitalized patients are most exposed to CDI. It is especially true for surgical patients who are taking antibiotics not only to treat infections but also as perioperative antimicrobial prophylaxis (PAP) for, among others, the prevention of surgical site infection (SSI). PAP should be administered for operative procedures that have a high rate of SSIs, or when foreign materials are implanted. An antimicrobial agent should be active against the pathogens most likely to contaminate the surgical site, given in an appropriate dosage and at an appropriate time. It is of crucial importance to administer the antibiotic for the shortest possible effective period, to minimize adverse effects, as is the case with CDI [9]. The choice of antibiotic depends primarily on the type of surgery.

Healthcare-associated CDI (HA-CDI) and community-associated CDI (CA-CDI) might differ, among others, due to other strains found in their respective environments. Exposure to antibiotic therapy also varies in hospital vs. the community. Although the binding of proton-pump inhibitors and nonsteroidal anti-inflammatory drugs to CDI is controversial, there are studies linking these aspects, especially with regards to CA-CDI [10]. In addition, a study by Na’amnih et al. linked IBD to an increased risk of developing CDI, which could be associated with altered host microbiota, as well as an increased permeability of the intestinal mucosa. However, interpretation of the results of this kind of research should be done cautiously [11].

The aim of the study was to analyze the epidemiology of CDI and antibiotic consumption in surgical patients of St Luke’s Provincial Hospital in Tarnow in 2012–2018, to determine the relationship between HA-CDI and antibiotic use and, secondly, to uncover a possible link between HAI-CDI/CA-CDI and seasonality.

## 2. Materials and Methods

### 2.1. Settings

Surveillance of CDIs in adult surgical wards was carried out at St Luke’s Provincial Hospital, one of the biggest teaching hospitals in southern Poland. The hospital comprises, among others, six surgical units—general, oncology, obstetrics, and gynecology (hospitalization of patients giving birth by caesarean section (surgical patients) as well as others (non-surgical patients)), neurosurgery, orthopedics and urology. This analysis only concerns the infections with CA-CDI and HA-CDI etiology, in patients of surgical wards. Results concerning other epidemiological data were already published [12].

### 2.2. Definitions

The data were collected in accordance with the methodology provided by the Healthcare-Associated Infections Surveillance Network (HAI-Net), European Centre for Disease Prevention and Control (ECDC). In Poland, the coordinator of HAI-Net was the Polish Society of Hospital Infections, a non-governmental organization. Participation in HAI-Net is voluntary and confidential. ECDC definitions [13] were used to diagnose new and recurrent infections. A case of CDI must meet at least one of the following criteria—(1) diarrheal stools or toxic megacolon and a positive laboratory assay for *C. difficile* toxin A or B in stools, or a toxin-producing *C. difficile* organism detected in stool via culture or other means, e.g., a positive PCR result; or (2) pseudomembranous colitis revealed by lower gastrointestinal endoscopy; or (3) colonic histopathology characteristic of *C. difficile* infection (with or without diarrhea) on a specimen obtained during endoscopy, colectomy, or autopsy. CA-CDI was defined as a case of CDI with the onset of symptoms (1) outside healthcare facilities and without discharge from a healthcare facility, within the previous 12 weeks, or (2) on the day of admission to a healthcare facility, or on the following day, and not being resident in a healthcare facility within the previous 12 weeks. HA-CDI was defined as a case of CDI with the onset of symptoms—(1) on day three or later, following admission to a healthcare facility on day one, or (2) in the community within four weeks of discharge from a healthcare facility (including the current hospital or a previous stay in any other healthcare facility). The analysis covered the cases of CA-CDI and HA-CDI that were diagnosed and classified during the patients’ stay in hospital, excluding recurrent cases of CDI. Recurrent CDI cases were defined as CDI cases with a positive *C. difficile* stool specimen between two to eight weeks of the last positive specimen [13]. All studied cases were from different patients, episodes of “recurrent” CDI (28 cases, 14.1% of all cases) were treated as an “outcome” and were not included in the study.

The basic scope of microbiological diagnostics of materials from patients with symptoms of gastrointestinal tract infection was established on the basis of our experience from previous years. All materials were tested for *Shigella* spp., *Salmonella* spp., rotavirus, norovirus, adenovirus, and *Clostridioides difficile*. CDI was confirmed with the use of C. DIFF QUIK CHEK COMPLETE, simultaneous detection of *C. difficile*-specific glutamate dehydrogenase (GDH) and toxins A and B, through enzyme immunoassay (TechLab, Blacksburg, VA, USA); cultures or PCR were not provided, and the ribotype was not determined [14]. The subjects qualified for the analysis were patients with both positive GDH and toxin results, without equivocal cases. No gastrointestinal tract infection of mixed etiology was identified.

### 2.3. Outcome Measurement Methodology

The following epidemiological measures were applied—the cumulative incidence and the density incidence. The incidence was calculated by dividing the number of CDI episodes by the number of patients admitted; incidence densities were calculated as the number of cases per number of patientdays × 10,000). The aggregate sum of the number of the defined daily doses (DDD) was in accordance with the ATC/DDD system of the World Health Organization (Anatomical Therapeutic Chemical classification system, group “J01”) [15]. Only antibiotics for systemic use were taken into account—no antifungal (J02), antimycobacterial (J04), or antiviral (J05) drugs were included in the analyses. The data referring to quantities of antibiotics used were expressed in the WHO-recommended defined daily doses (DDDs) [16]. The results of the analysis of the use of antibiotics were demonstrated using the DDD/1000 patient-days index. The total pool of antibacterials applied for systemic use (J01) was dominated by medicaments from only two groups—they constituted 67.5% of the total—and they were J01D (other β-lactam antibiotics—cephalosporins, monobactams, carbapenems, and penems); and J01M (fluoroquinolones). Hence, detailed data concerning antibiotic consumption were limited to the drugs indicated.

In the course of the study, uniform principles of perioperative antibiotic prophylaxis were in force in the hospital, which were applied, depending on the type of surgery either during the stay in the operating theatre only or also in the postoperative period for 48 or 72 h (Appendix A). In the study period, all patients underwent the procedure described in accordance with the principles given regarding the choice of antibiotics, route and length of administration (the procedure was not validated).

### 2.4. Statistical Analysis

For ordinal and nominal variables, Pearson’s chi-square test (χ2) was used. Linear regression was carried out, in which the dependent (explained) variable was CDI incidence/10,000 patientdays (pds) and the independent (explanatory) variable was antibiotic use in DDD/1000 pds; the analysis was done with the antibiotics divided into groups: J01, J01M, and J01D. For statistical analysis of ordinal numbers or dichotomous data, information on the number and percentage of people was used. The average median (Me), 95% confidence interval (95% CI), minimum and maximum were calculated. Statistical significance was assumed at *p* < 0.05. Statistical analysis of the data was carried out using the SPSS software (SPSS–Statistical Package for the Social Sciences, STATISTICS 24, Armonk, NY, USA).

### 2.5. Ethics Approval and Consent to Participate

The use of data was approved by the Bioethics Committee of the Jagiellonian University (No. KBET 122.6120.118.2016 25.05.2016). All data were anonymized.

## 3. Results

In the study period, there was a total of 69,580 patients in the surgical wards, the majority of whom were hospitalized in the obstetrics and gynecology (23.6% of the total) and general surgery wards (23.1% of the total). The average length of stay (LOS) was significantly dependent on the type of ward (*p* < 0.0001) and amounted to 4.8 days, with the shortest LOS of 3.9 days in obstetrics and gynecology (SD = 4.4 days) and urology, (SD = 6.8 days), and the longest of 6.2 days in the orthopedic ward (SD = 5.0 days) (Table 1). There was a statistically significant difference between the days of HA-CDI onset in different surgical patients (F = 4.92, *p* < 0.0001; Table 1).

In total, 197 cases of CDI were identified, of which 147 (72.4%) were HA-CDI. The most CA-CDI cases were observed in the general surgery (32.6%) and urology (17%) wards (15 cases each), the incidence of CA-CDI was 0.7/1000 patients. HA-CDI was diagnosed, on average, on the 14th day of stay and its incidence was 2/1000 patients (4.4/10,000 pds). The highest percentage of HA-CDI among all CDI cases was observed in urology, where 83.1% of all cases were HA-CDI, also the highest HA-CDI incidence was noted in the urology department—0.7% (18.0/10000 pds) (Table 1).

A significant positive correlation in all departments was found between CA-CDI and HA-CDI (*r* = 0.943, *p* < 0.001) and between the number of patients and HA-CDI (*r* = 0.865, *p* = 0.012).

The peak of HA-CDI incidence was observed in autumn and winter, from October to December, and additionally from May to July. There was no significant relationship between the occurrence of CA-CDI in the individual months of the year and HA-CDI, (Phi = 17.968, *p* = 0.082, Figure 1).

During the study period, antibiotic consumption in all wards reached 244,395 DDD and 0.18 DDD/1000 pds; it was the lowest in the neurosurgery ward (0.46 DDD/1000 pds) and the highest in the urology department (0.84 DDD/1000 pds). The highest proportion of fluoroquinolones was observed (group J01M) in the urology unit, which constituted 49.5% of all antibiotics, while their average share in the wards under study was 17.8%.

Linear regression for the incidence density of HA-CDI (dependent variable) and the total antibiotic consumption J01 DDD/1000 pds (independent variable) did not show a straight-line relationship (F = 5.44; *p* = 0.080; R^2^ = 0.47; Table 2). Similarly, no rectilinear relationship was found in the analysis of beta-lactams J01D (F = 0.079; *p* = 0.793; R^2^ = 0.01). For fluoroquinolones J01M, the result of the linear regression showed a good fit of the model (F = 42.94, *p* = 0.003), and the regression coefficient confirmed the fact that the use of antibiotics from the group J01M was strongly positively associated with the HA-CDI incidence density (R^2^ = 0.915; Table 2).

## 4. Discussion

In total, during the study period, a significant proportion of CDI cases were HA-CDI cases, while the CA-CDI cases accounted for approximately one-fourth of the total. This situation was typical and was confirmed by other researchers [10]. In fact, our observation that HA-CDIs outnumbered CA-CDIs in surgical patients might not be surprising, as with the exception of general surgery, one would not expect CA-CDI or any other common infection to be admitted to neurosurgical or obstetrical wards. Factors that predispose the development of HA-CDI might be divided into two general categories—host factors and factors that disrupt normal colonic microbiota (antibiotics or other medications and surgery). Colorectal surgery (with the combination of cefazolin with or without metronidazole for PAP [10]), in cancer patients is, among others, a documented risk factor for CDI. According to Aquina et al., the incidence of CDI after colorectal resection was 2.2% [15] and according to Yeon, it was 6.8% [17]. In our population, it was about 0.3%. In a multicenter study in the USA, postoperative CDI occurred in 0.02% of patients treated with radical prostatectomy (with fluoroquinolones for PAP [10]), 0.23% of those treated with partial or radical nephrectomy and 1.7% of those treated with radical cystectomy [18]. In our study, this percentage was 0.7%. The risk of HA-CDI in critically injured trauma patients is also high; according to Lumpkins et al., the incidence is 3.3% [19]. The incidence of HA-CDI in patients undergoing other types of surgery (with cefazolin for PAP [10]) is lower, in general surgery (cholecystectomy and appendectomy)—0.5% [20], lumbar spine surgery—0.1%, [21], and in caesarean section or gynecological procedures it is 0.1–0.2% [22,23]. The present data indicate similar relationships.

The urology ward is an exception since the incidence here differs significantly from expectations. In this unit, a very high total consumption of antibiotics significantly affected the CDI incidence. These were especially fluoroquinolones, which constituted half of all antibiotics, and fluoroquinolones were also additionally used in the perioperative antibiotic prophylaxis regimen. Additionally, while the administration of fluoroquinolones in urology during the perioperative antibiotic prophylaxis was consistent with the literature on the subject, prolonging their use beyond the perioperative period, i.e., administration of antibiotics outside the operating room for several days was very controversial [24].

The observed situation of high HA-CDI incidence with increasing duration of antimicrobial prophylaxis was also reported by other authors. In cardiac surgery, in orthopedic total joint replacement, as well as colorectal and vascular procedures, higher odds of CDI were significant in a duration-dependent fashion; on the other hand, extended duration did not lead to additional SSI reduction [25]. Additionally, other indications confirmed our hypothesis that the most critical risk factor for CDI was the use of antibiotics, which disrupted the natural intestinal microbiota [26], and according to the literature data, the antibiotics that exhibited the greatest risk of being conducive to the development of *C. difficile* infections included fluoroquinolones, which were often used in the ward under study and which increased the risk of developing CDI five-fold [27]. Our studies have confirmed this relationship indirectly since the high consumption of fluoroquinolones in the urology ward was associated with a high incidence of CDI. Fortunately, experiences of other authors demonstrated that it is possible to reverse the trend. According to Turner et al., following the implementation of a multidisciplinary intervention, including reductions in cefoxitin and fluoroquinolone use, showed a decrease in CDI incidence from 1.27% in 2016 to 0.91% in 2017 was obtained [28].

On the other hand, the present data were also confirmed by data on CDI epidemiology in Europe as, according to the 2016 ECDC report, the most common CD ribotype, both in Europe and in Poland, was RT027 [6,29]. Unfortunately, epidemic ribotype RT027 is associated with multiple antimicrobial resistance, also to fluoroquinolones [30]. Perhaps, the present data also explain the phenomenon described by Pituch et al., concerning the very high proportion of ribotype RT027 in Poland, which was over two-fold higher, 23% vs. 48%, compared to other European data [29]. Currently, fluoroquinolones remain one of the most frequently administered drugs in the treatment of Polish patients and their proportion in outpatient treatment grew significantly from 2007 to 2016 and amounted to 6% of the total antibiotic sales [31].

Unfortunately, no relationship was found between the growth of the number of CA-CDI and HA-CDI in the second of the two periods of increase in the number of cases, i.e., in spring and summer. The number of HA-CDI cases grew regardless of the small numbers of CA-CDI. It was particularly applicable to the urology ward, in which the highest total incidence of HA-CDI was demonstrated. Therefore, it is not a surprise that the dynamics of the occurrence of CA-CDI were variable, as the growth in the number of cases of CDI in winter was probably associated with the period of colds and an increased use of antibiotics in the general population [32]. In the study by Gilca et al., infection with the influenza virus and RSV (respiratory syncytial virus) affected the CDI, regardless of the antibiotic prescribed. Similarly, in a study by Barrett et al., a relationship was recorded between an increased incidence of norovirus and CDI [33]. Additional research is necessary to assess the degree to which these determinants contribute to the seasonality of CDI.

It is probable that the high incidence of HA-CDI was also a reflection of a trend characteristic of Polish hospitals. According to ECDC data, Polish hospitals boast one of the highest CDI incidence rates. In 2016, in Poland, the median was 1.4 cases/10,000 pds, while for Europe it was only 0.8 cases/10,000 pds [6]. Sadly, this study confirmed these data. According to Krutova et al., epidemiology of CDI could be used as an indicator of healthcare quality at hospital, as well as at the country level [34].

An infection control “bundle” strategy should be used to successfully reduce HA-CDI [35]. The “bundle” approach should include multifaceted interventions, including antibiotic stewardship, hand hygiene with soap and water, isolation measures, including contact precautions for patients with CDI, and environmental hygiene. For surgery patients, it is very important to also reduce environmental exposure to CDI through timelier preoperative medical optimization in the outpatient setting [36] and to apply perioperative antibiotic prophylaxis that employs the most optimal regimen for the patient.

Limitations of this study were that it lacked knowledge on (i) the strain characteristics, i.e., their drug resistance, which might be of importance in determining the level of fluoroquinolone consumption and the high HA-CDI incidence in the urology ward; (ii) data on the medical history of patients regarding their outpatient use of antibiotics before hospitalization; and in addition, (iii) the ribotype was not determined. Only antibiotics for systemic use were taken into account in our study, excluding antifungal, antimycobacterial, and antiviral. There are no sufficient reasons to conclude that there was a close relationship between HA-CDI incidence and consumption of these drugs. Only single reports on this subject were found, e.g., regarding the use of valaciclovir [37] or voriconazole [38].

## 5. Conclusions

Not only antimicrobial therapy, but also perioperative antibiotic prophylaxis schemes might have a significant impact on the CDI incidence.

## Figures and Tables

**Figure 1 microorganisms-08-00810-f001:**
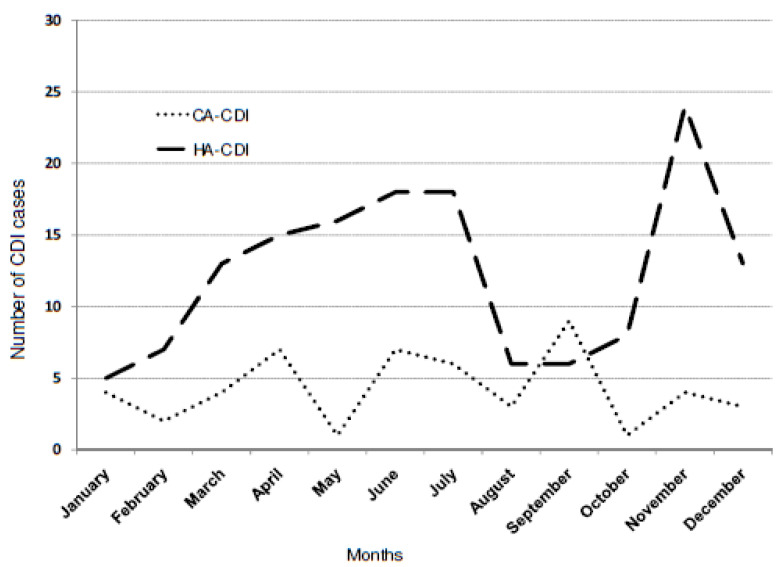
The occurrence of community-acquired CDI (CA-CDI defined by the ECDC) and the hospital-acquired CDI (HA-CDI, defined by the ECDC) in 2012–2018, at St. Luke Hospital in Tarnów, Poland.

**Table 1 microorganisms-08-00810-t001:** Epidemiology of *Clostridioides*
*difficile* infections (CDI) in the surgical wards at St Luke’s Provincial Hospital, Tarnow, in 2012–2018.

Hospital Ward	No of Patients	No. of CDI Cases (*n*)		Length of Stay (days)	HA-CDIs in All CDIs (%)	Day of HA-CDI Onset (SD)	Incidence of HA-CDI		OR 95%CI
		HA	CA				Per 100 Admissions	Per 10,000 pds	
General surgery	16,057	31	15	5.01	28.6	14 (9.9)	0.2	3.9	7.9 (2.79–22.43)
Oncology	4643	16	7	4.09	76.9	20 (15.5)	0.3	8.4	14.1 (4.72–42.23)
Obstetrics and gynecology	16,398	4	10	3.97	92.3	9 (4.4)	0.0	0.6	ref
Neurosurgery	12,936	10	3	5.34	67.4	26 (29.8)	0.1	1.4	3.2 (0.99–10.11)
Orthopedics	9373	11	1	6.17	69.6	10 (5.0)	0.1	1.9	4.8 (1.53–15.11)
Urology	10,173	73	15	3.98	83.1	11 (6.8)	0.7	18.0	29.4 (10.75–80.51)
Total/average	69,580	147	51	4.77	74.2	14 (12.9)	0.2	4.4	n/a

HA—Healthcare-associated, CA—community-associated, HA-CDI—Healthcare-associated–CDI, SD—standard deviation, and OR—odds ratio.

**Table 2 microorganisms-08-00810-t002:** Antibiotic consumption in the surgical wards at St Luke’s Provincial Hospital, Tarnow, in 2012–2018, and the correlation of total consumption and consumption of selected groups of antibiotics and HA-CDI incidence density.

Hospital Ward	Antibiotics J01		Antibiotics J01M		Antibiotics J01D	
	DDD	DDD Per 1000 pds	DDD	DDD Per 1000 pds	DDD	DDD Per 1000pds
Obstetrics and gynecology	49,219	0.66	137	0.00	41,649	0.56
Neurosurgery	36,566	0.46	6485	0.08	19,678	0.24
Orthopedics	35,070	0.51	1591	0.02	14,841	0.22
General surgery	66,329	0.72	13,352	0.15	23,022	0.25
Oncology	18,320	0.76	2708	0.11	8322	0.34
Urology	38,891	0.84	19,247	0.41	13,972	0.30
Total/average	244,395	0.18	43,520	0.03	121,484	0.09
average (SD]	n/a	0.7 (0.15)	n/a	0.1 (0.15)	n/a	0.3 (0.16)
Linear regression, correlation HA-CDI incidence density per 10.000 pds vs. DDD per 1000pds						
Beta	0.759		0.956		−0.139	
*p*-value	0.080		0.003		0.793	

DDD Consumption of antibacterials for systemic use (J01) expressed in defined daily doses per 1000 patientdays (pds); J01 antibacterials for systemic use; J01D other β-lactam antibiotics—first, second, third, and fourth-generation cephalosporins, monobactams, carbapenem, and other cephalosporins and penems; J01M fluoroquinolones; n/a—not available; Pds—patientdays

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
