# Peer review of "Long-Term Antibiotic Prophylaxis in Urology and High Incidence of Clostridioides difficile Infections in Surgical Adult Patients"

_microorganisms, 2020, doi:10.3390/microorganisms8060810_

Round 1
Reviewer 1 Report
The manuscript entitled „LONG-TERM ANTIBIOTIC PROPHYLAXIS IN UROLOGY AND HIGH INCIDENCE OF CLOSTRIDIOIDES DIFFICILE INFECTIONS IN SURGICAL ADULT PATIENTS” by Jachowicz et al. is an interesting and current paper describing the effects of prophylactic antibiotic use on the incidence of C. difficile. The paper is definetly of interest to the Journal and may expect a high readership.
The paper uses appropriate methods, is concisely-written, and uses good English.
However, there are some concerns that need to be addressed before the paper could be accepted for publication:
- The paper does not follow the instructions for authors, this must be corrected.
Introduction:
What is the significance of the yellow and blue highlight? This must be removed.
L53: C. difficile should be written in full on the first mention in the body of text, and later discussed in only its abbreviated form (C. difficile)
The authors should at least discuss the taxonomic change of C. difficile (Clostridium or Clostridioides) of this pathogen, even if they choose not to use this taxonomic designation consistently. Some parts of the papers discusses Clostiridium, while others Clostridioides.
Please include the following reference:
https://doi.org/10.1016/j.anaerobe.2016.06.008
L53: for modern medicine and healthcare infrastructures worldwide.
L57: please discuss in a few sentences the emerging role of C. difficile in extra-intestinal infections, please include the following reference:
Antibiotics 2020, 9(1), 16; https://doi.org/10.3390/antibiotics9010016
Materials and methods:
Subsections should be used!
L138: Please include the following reference:
Antibiotics 2017, 6(4), 25; https://doi.org/10.3390/antibiotics6040025
L146-L148: although these drugs were not taken into consideration, please say a few sentences in the discussion whether there has been studies showing the relevance of these agents in inducing or exacerbating C. difficile infections.
Author Response
DETAILED RESPONSE TO REVIEWERS:
STEP-BY-STEP REPLIES TOREVIEWERS' COMMENTS:
Reviewer #1:
The manuscript entitled „LONG-TERM ANTIBIOTIC PROPHYLAXIS IN UROLOGY AND HIGH INCIDENCE OF CLOSTRIDIOIDES DIFFICILE INFECTIONS IN SURGICAL ADULT PATIENTS” by Jachowicz et al. is an interesting and current paper describing the effects of prophylactic antibiotic use on the incidence of C. difficile. The paper is definitely of interest to the Journal and may expect a high readership.
The paper uses appropriate methods, is concisely-written, and uses good English.
Authors’ reply: Thank you for this comment!
The paper does not follow the instructions for authors, this must be corrected. What is the significance of the yellow and blue highlight? This must be removed.
Authors’ reply: The manuscript was after 1st round of review, “yellow and blue highlight” was a designation of changes in the text: they are easily visible to the editors and reviewers. But currently, the previous mark has been removed.
L53: C. difficile should be written in full on the first mention in the body of text, and later discussed in only its abbreviated form (C. difficile)
Authors’ reply: Corrected according to suggestions.
The authors should at least discuss the taxonomic change of C. difficile (Clostridium or Clostridioides) of this pathogen, even if they choose not to use this taxonomic designation consistently. Some parts of the papers discusses Clostridium, while others Clostridioides. Please include the following reference: https://doi.org/10.1016/j.anaerobe.2016.06.008
Authors’ reply: Corrected according to suggestions (lines 58-64), as below:
“(…) The limitation of the genus Clostridium to Clostridium butyricum and related species means that a lot of existing microorganisms shouldn’t be considered Clostridium sensu stricto. One such example of medical significance is C. difficile. Based on 16S rRNA gene sequence analysis, the closest relative of Clostridium difficile is Clostridium mangenotii, and both are in the Peptostreptococcaceae family, which is phylogenetically far from members of Clostridium sensu stricto. Based on phenotypic, chemotaxonomic and phylogenetic analyses, a new type proposed for Clostridium difficile is Clostridioides difficile [2]. (…)”
L53: for modern medicine and healthcare infrastructures worldwide.
Authors’ reply: Corrected according to suggestions.
L57: please discuss in a few sentences the emerging role of C. difficile in extra-intestinal infections, please include the following reference: Antibiotics 2020, 9(1), 16; https://doi.org/10.3390/antibiotics9010016
Authors’ reply: Corrected according to suggestions (lines 75-82), as below:
“(…) Parenteral CDI are very rare and are characterized by high mortality. In a study conducted by Urbán et al. 4 129 CD strains were isolated; among which only 24 (0.58%) isolates came from parenteral sources. Eight strains were isolated from abdominal infection: appendicitis, rectal abscess or from people with Crohn's disease. Patients did not have diarrhoea. In most cases, CD was obtained as a part of the mixed bacterial flora. The isolates were most often capable of producing toxins and belonged mainly to the PCR 027 ribotype. There is an increasing number of reports of C. difficile-related parenteral infections [8] (…)”
Materials and methods: Subsections should be used!
Authors’ reply: Corrected according to suggestions.
L138: Please include the following reference:
Antibiotics 2017, 6(4), 25; https://doi.org/10.3390/antibiotics6040025
Authors’ reply: Corrected according to suggestions.
L146-L148: although these drugs were not taken into consideration, please say a few sentences in the discussion whether there has been studies showing the relevance of these agents in inducing or exacerbating C. difficile infections.
Authors’ reply: Corrected according to suggestions (lines 309-313), as below:
“(…) Only antibiotics for systemic use were taken into account in our study, excluding antifungal, antimycobacterial and antiviral. there are no sufficient reasons to conclude that there is a close relationship between HA-CDI incidence and consumption of these drugs. Only single reports on this subject are found, e.g. regarding the use of valaciclovir [38] or voriconazole [39]. (…)”
Reviewer 2 Report
CORRECTED MANUSCRIPT ID: microorganisms-707283
Comments
Introduction. Background data contain specific pathogenicity on cellular level that are interesting but out of the scope of this ms and could easily be cut down. More appropriate background parts are indicated in yellow and well written.
Page 3. Line 54-55. CDI is becoming “harder and more expensive to treat”. A statement general in character without reference. Please clarify for better understanding.
Line 74-76. This statement is not correct and misleading nor can it be supported in the references. Treatment failure is not 25% since usually the patient respond to therapy but within 30 days experience a recurrence with this frequency. The fraction of relapse is impossible to estimate even with modern WGS technique due to common environmental contamination. Revision is needed.
Page 5. Line 129-131. Recurrent episode has to be defined (within 60 days of primary episode?) and the size of this exclusion should be mentioned (25%?)
Line 135-137. Diagnostic test in use is obscure. What is the trade mark:” commercial Clostridioides difficile test cassette (TechLab, Blacksburg, VA, USA): there was simultaneous detection of GDH and toxins A and B by enzyme immunoassay” Please clarify.
Page 6. Line 156. “perinatal”?????
Results
Page7. Line 180-184. SD does not need to be in the text. Please revise.
Line 191-193. These significant associations is not clear to me. CA-CDI correlates with HA-CDI for all departments? What does this mean and whats the rationale? The odd association with no of patients and HA-CDI (regardless of ab use?) is also strange?
Discussion
Page 8. Line 211-216. First, bad English when “concerned” is used frequently in wrong context. Line 213-16. Sentence is totally unclear to me. Rephrase..
Line 218-28. Comparison of risk of different surgery and CDI is worth little without relating to perioperative antibiotics. Please add data on similar ab use in referred studies and refer to supplement?
Page 10. Line 289-296. This paragraph (correction) is not clear and what´s the message? What diagnostic method was used in the study compared to the ESCNID recommendation?
Line 300-307. Conclusion should include main results from the urology and other departments from the study period and antibiotic consumption. Compare with abstract and keep a “red line”. I find the seasonal variations of little interest.
Author Response
DETAILED RESPONSE TO REVIEWERS:
STEP-BY-STEP REPLIES TO REVIEWERS' COMMENTS:
Reviewer #2:
Introduction. Background data contain specific pathogenicity on cellular level that are interesting but out of the scope of this MS and could easily be cut down. More appropriate background parts are indicated in yellow and well written.
Authors’ reply: Thank you for this comment! Corrected according to suggestions.
Page 3. Line 54-55. CDI is becoming “harder and more expensive to treat”. A statement general in character without reference. Please clarify for better understanding.
Authors’ reply: Corrected according to suggestions, the paragraph has been redrafted
Line 74-76. This statement is not correct and misleading nor can it be supported in the references. Treatment failure is not 25% since usually the patient respond to therapy but within 30 days experience a recurrence with this frequency. The fraction of relapse is impossible to estimate even with modern WGS technique due to common environmental contamination. Revision is needed.
Authors’ reply: Corrected according to suggestions (lines 71-74), as below:
“(…) Empirical therapy for CDI is effective as regards approx. 75% of cases, while 25% of patients with CDI do not respond to antibiotic therapy, and additionally, in approx. 8% if cases there are relapses. However, the fraction of relapse is impossible to estimate even with modern the whole genome sequencing techniques due to common environmental contamination [5, 6]. (…)”
Page 5. Line 129-131. Recurrent episode has to be defined (within 60 days of primary episode?) and the size of this exclusion should be mentioned (25%?)
Authors’ reply: Corrected according to suggestions (lines 138-142), as below:
“(…) The analysis covered the cases of CA-CDI and HA-CDI that were diagnosed and classified during the patients’ stay in hospital, excluding recurrent cases of CDI. Recurrent CDI cases was defined as CDI cases with a positive C. difficile stool specimen between two to eight weeks of the last positive specimen [13]. All the studied cases were from different patients, episodes of “recurrent” CDI (28 cases, 14.1% of all) were treated as an “outcome” and were not included in the study. (…)"
Line 135-137. Diagnostic test in use is obscure. What is the trade mark:” commercial Clostridioides difficile test cassette (TechLab, Blacksburg, VA, USA): there was simultaneous detection of GDH and toxins A and B by enzyme immunoassay” Please clarify.
Authors’ reply: Corrected according to suggestions (lines 146-148), as below:
“(…) CDI was confirmed with the use of C. DIFF QUIK CHEK COMPLETE, simultaneous detection of GDH and toxins A and B by enzyme immunoassay (TechLab, Blacksburg, VA, USA) (…)”
Page 6. Line 156. “perinatal”?????
Authors’ reply: Corrected according to suggestions, I am sorry for my mistake!.
Results, Page7. Line 180-184. SD does not need to be in the text. Please revise.
Authors’ reply: Corrected according to suggestions.
Line 191-193. These significant associations is not clear to me. CA-CDI correlates with HA-CDI for all departments? What does this mean and what’s the rationale? The odd association with no of patients and HA-CDI (regardless of ab use?) is also strange?
Authors’ reply: Corrected according to suggestions (lines 203-208), as below:
“(…)A significant positive correlation in all departments was found between CA-CDI and HA-CDI (r=0.943, p<0.001) and between the number of patients and HA-CDI (r=0.865, p=0.012). (…)”
Discussion, Page 8. Line 211-216. First, bad English when “concerned” is used frequently in wrong context.
Authors’ reply: Corrected according to suggestions.
Line 213-16. Sentence is totally unclear to me. Rephrase.
Authors’ reply: Corrected according to suggestions (lines 227-232)
Line 218-28. Comparison of risk of different surgery and CDI is worth little without relating to perioperative antibiotics. Please add data on similar ab use in referred studies and refer to supplement?
Authors’ reply: Corrected according to suggestions (lines 234, 239, 243).
Page 10. Line 289-296. This paragraph (correction) is not clear and what´s the message? What diagnostic method was used in the study compared to the ESCMID recommendation?
Authors’ reply: Corrected according to suggestions, the paragraph was canceled.
Line 300-307. Conclusion should include main results from the urology and other departments from the study period and antibiotic consumption. Compare with abstract and keep a “red line”. I find the seasonal variations of little interest.
Authors’ reply: Thank you for this comment! Corrected according to suggestions.
This manuscript is a resubmission of an earlier submission. The following is a list of the peer review reports and author responses from that submission.
Round 1
Reviewer 1 Report
intersting epidimiology work that can contribute to the community of physicians working in the field
Reviewer 2 Report
This is a descriptive study regarding the epidemiology of C. difficile infections in surgical wards of a Polish hospital from 2012-2018.
Authors' observation, that hospital-associated CDIs outnumber community associated CDIs in surgical wards might not be surprising as with the exception of general surgery, one would not expect community associated CDI to be admitted in neurosurgery or obstetrical wards.
The fact that antibiotic consumption is related to increase in CDI rates is well established. Comparisons between different surgical specialties without taking into account the different population characteristics is troublesome.
The authors have a valid concern that the increasing duration of peri-operative prophylactic antibiotics can be related to increasing rates of CDI. They do little to address how many patients received long perioperative prophylaxis and if this was associated with CDI.
The manuscript might benefit from a statistical and English review. The references need to be double checked as in various occasions the information provided is not supported by the reference.
Reviewer 3 Report
In this article the authors describe the incidence and epidemiology of CDI in a provincial hospital from Poland during seven years. The authors distinguish cases of HA- and CA-CDI, and they try to analyze the relation between antibiotic consumption and CDI in their patients. At the outset, these objectives could be considered interesting in order to know their local epidemiology as well as analyze their antibiotic consumption policy and the relationship with CDI. In addition, the main conclusion of the work relating the antibiotic prophylaxis that is applied to urological patients with C. difficile infection, in principle, is interesting.
However, some major comments should be considered:
- As the authors themselves comment, there are great limitations in this work, since the molecular epidemiology and the antibiotic sensitivity of the isolates are not analyzed; being Poland a country with a very high incidence of difficile ribotype 027 that is associated with a high resistance rate to fluoroquinolones. Therefore, they could safely demonstrate that the high incidence in these patients would be due to the use of fluoroquinolones.
- The laboratory technique for the study of difficile has a low sensitivity to be used as a single diagnostic method, so that, probably, the actual cases are more than those diagnosed and, therefore, the incidence data is not real. This means that there is a bias in the detection of cases and therefore, it is a further limitation of work.
- The manuscript is not drafted clearly and they also need to review the english.
- The summary should be written more clearly and the conclusion mentioned in the title should be commented.
- The introduction should better introduce the topics discussed at work: hospital and community CDI, relationship of antimicrobial prophylaxis with infection and types of antibiotics…
Minnor comments:
- CA definition is not included at any time.
- Lin37: % of HA-DCIs is missing
- Lin73-75: maybe the authors should rewrite the paragraph since it is not well expressed.
- Lin 86: the hospital comprises nine or six surgical units?? the authors list six.
- Lin 101: without instead withveut
- Lin 111: few enteropathogens are studied in patients with gastrointestinal symptoms. Can the authors explain why only those listed are studied?
- Lin 154: are the all de the cases from different patients or different episodes?
- Lin 164: the description of results does not match the figure. Maybe CA -and HA-CDI are exchanged in the figure.